# Signatures of spinorial order in $URu_2Si_2$: Landau-Ginzburg theory of hastatic order

Milan Kornjača[1], Rebecca Flint[1*],

**1** Department of Physics and Astronomy, Iowa State University, 12 Physics Hall, Ames, Iowa 50011, USA

* flint@iastate.edu

July 12, 2021

## Abstract

The hidden order in $URu_2Si_2$ remains a compelling mystery after more than thirty years, with the order parameter still unidentified. One intriguing proposal for the phase has been hastatic order: a symmetry breaking heavy Fermi liquid with a spinorial hybridization that breaks *both* single and double time-reversal symmetry. Hastatic order is the first *spinorial*, rather than vectorial order in materials, but previous work has not yet found direct consequences of the spinorial nature. In this paper, we revisit the hastatic proposal within Landau-Ginzburg theory. Rather than a single spinorial order parameter breaking double-time-reversal symmetry, we find two gauge invariant vectorial orders: the expected composite order with on-site moments, and a new quantity capturing symmetries broken solely by the spinorial nature. We address the effect of fluctuations and disorder on the tetragonal symmetry breaking, explaining the absence of in-plane moments in $URu_2Si_2$ and predicting a new transition in transverse field.

# 1  Introduction

URu$_2$Si$_2$ is an Ising heavy fermion material that undergoes a phase transition into an unknown state, known as "hidden order" (HO) at $T_{HO} = 17.5$K [1–8]. While the large entropy at the transition suggests a large order parameter [1], no large moments have ever been found. Translation symmetry is clearly broken [9–11], but other broken symmetries [12, 13], particularly tetragonal symmetry are more controversial [14–19], with apparently conflicting results. The problem has given rise to a number of fascinating theoretical proposals [13, 20–32], in part driven by the difficulty in determining the appropriate underlying microscopic model in actinide materials.

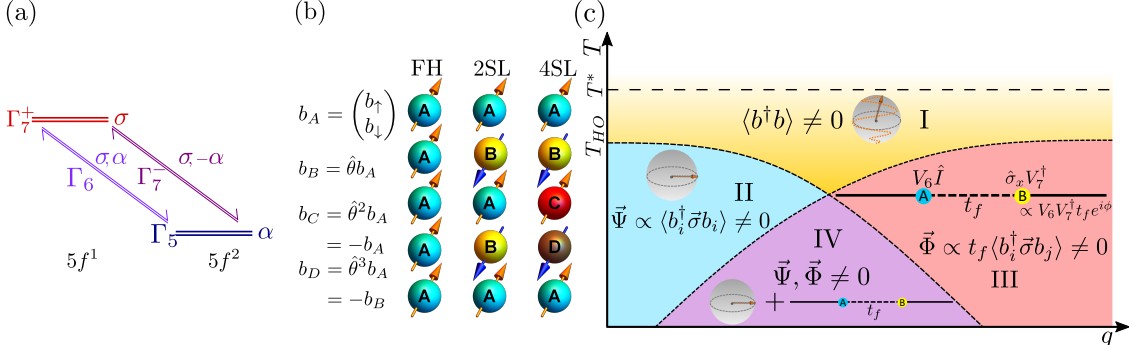

Figure 1: (a) Hastatic order arises from valence fluctuations between the $U^{4+}$ ground state $\Gamma_5$ non-Kramers doublet and an excited $\Gamma_7^+$ Kramers doublet that lead to two-channel Kondo physics, in which the finite excited state occupation ($b_\sigma$) breaks $SU(2)$ channel symmetry. (b) As the order parameter is fundamentally a spinor, distinct spinor arrangements (color) break different symmetries, even with identical moment structure (arrows). This can be seen in the two and four sublattice antiferrohastatic phases, which break time-reversal and inversion symmetries, respectively. (c) Generic phase diagram of antiferrohastatic order beyond mean-field theory. The spinor amplitude can onset gradually with a coherence temperature $T^*$, followed by second order transitions for the two possible order parameters, $\vec{\Psi}$ representing the moment ordering and $\vec{\Phi}$ representing RKKY hopping induced inter-sublattice correlations, with three ordered phases: $\vec{\Psi}$ only ($II$), $\vec{\Phi}$ only ($III$) and mixed ($IV$).

Hastatic order was proposed to explain the HO as a symmetry breaking heavy Fermi liquid, where the order parameter is the hybridization gap itself, and the moments are nat-

urally suppressed by $T_{HO}/D$, where $D$ is the conduction electron bandwidth [31,33]. This hybridization is generated by valence fluctuations from a $\Gamma_5$ non-Kramers doublet ground state to an excited Kramers doublet. The order parameter may be treated as the condensation of a *spinor* of auxiliary bosons, $\langle b_{j\sigma} \rangle$ representing the excited state occupation. The amplitude, $\langle b^\dagger b \rangle$ gives an Ising anisotropic heavy Fermi liquid, capturing hybridization gaps [5–8,34] and heavy masses [1], as well as the Ising anisotropic Fermi surface magnetization [3,11,35,36] and non-linear susceptibility [4]. The direction, $\langle b^\dagger \vec{\sigma} b \rangle$ breaks symmetries, and the staggered basal plane order of these excited moments is consistent with the physics of HO. Several key open questions remain: experimentally, hastatic order predicts tiny transverse moments, $\langle b^\dagger \vec{\sigma}_\perp b \rangle$ that have not been found in neutrons [37–39], as well as an associated broken tetragonal symmetry that remains experimentally controversial [14–19,40]; theoretically, hastatic order would be the first *spinorial* order in materials, yet thus far all predicted signatures arise from the *vectorial* composite order parameter, $\langle b^\dagger \vec{\sigma} b \rangle$.

We resolve these questions by developing the Landau-Ginzburg theory for tetragonal hastatic order with Landau parameters motivated by microscopic theory [31,41]. In addition to the *on-site* moments, $\langle b_i^\dagger \vec{\sigma} b_i \rangle$, the spinorial order generically requires a second order parameter associated with *intersite* interference, $\sim \langle b_i^\dagger \vec{\sigma} b_j \rangle$ that captures the double time-reversal symmetry breaking. We revisit the tetragonal symmetry breaking to find three relevant couplings with vastly different microscopic magnitudes that reconcile the experimental literature, and show that weak in-plane anisotropy leads to soft transverse fluctuations that couple linearly to random strains. These lead to an Imry-Ma-like loss of tetragonal symmetry breaking beyond a small critical disorder that explains why neutrons find no basal plane moments, even as $\mu$SR and NMR see disordered local fields [42,43]. The fluctuations can be stiffened by external strain or transverse magnetic fields, predicting an additional ordering transition where the transverse moments develop that should be observable in neutrons or elastoresistivity under applied strain or large transverse fields.

This paper is organized as follows: we review the microscopic model and introduce the hastatic spinor in Sec. 2, and construct three gauge-independent quantities that capture the symmetries broken by hastatic order in Sec. 3. Broken symmetries and moments are further explored in Sec. 4 and used to derive an effective antiferrohastatic Landau theory in Sec. 5, where we focus on distinct signatures of tetragonal symmetry breaking and novel transverse field transitions. In Sec. 6, we examine the effect of fluctuations and disorder on tetragonal symmetry breaking before concluding in Sec. 7.

## 2 Microscopic model

The simplest microscopic model for tetragonal hastatic order is the infinite-$U$ two-channel Anderson model depicted in Fig. 1(a), where local $\Gamma_5$ non-Kramers moments fluctuate to an excited Kramers doublet ($\Gamma_7^+$) via two channels of conduction electrons that lie in different tetragonal irreducible representations, $\Gamma_6$ and $\Gamma_7^-$. As we focus on the phenomenology, we take a schematic version of this Hamiltonian that captures the essential details,

$$H = \sum_{\mathbf{k}} \epsilon_{\mathbf{k}} c_{\mathbf{k}\sigma\alpha}^\dagger c_{\mathbf{k}\sigma\alpha} + \Delta E \sum_j b_{j\sigma}^\dagger b_{j\sigma} + \sum_{ij} t_{f,ij} f_{i\alpha}^\dagger f_{j\alpha} + \sum_j c_{j\sigma\alpha}^\dagger b_{j\sigma}^\dagger \left( V_6 f_{j\alpha} + V_7 f_{j-\alpha} \right) + H.c.$$

(1)

The first term describes two bands of non-interacting conduction electrons ($c_{\mathbf{k}\sigma\alpha}$) that can mix with the $\Gamma_5$ non-Kramers doublet ground state of 5f$^2$ $U^{4+}$ ($f_{j\alpha}$) via valence fluctuations to an excited Kramers doublet (5f$^1$ or 5f$^3$) at energy $\Delta E$ ($b_{j\sigma}$). The original infinite-U

model is written in terms of Hubbard operators, which we have already eliminated by introducing this representation of the excited Kramers doublet by a auxiliary boson $b_{j\sigma}$, and the non-Kramers ground state by a auxiliary fermion, $f_{j\sigma}$ as derived in Ref. [[31]]. The development of hastatic order is represented by the condensation of the auxiliary bosons, which leads to the development of hybridization gaps and heavy Fermi liquid physics, in addition to the symmetry breaking aspects discussed in this work. This mean-field is exact in the large-$N$ limit where the $SU(2)$ pseudospin of the ground state doublet ($\alpha$) is generalized to $SU(N)$. Tetragonal symmetry leads to two conduction electron channels ($\sigma$) that hybridize via two different symmetries $\Gamma_6$ and $\Gamma_7^-$. We have included an "f-electron hopping" term, $t_f$ that moves auxiliary fermions between sites, but it is important to note that this is not the bare hopping of the original f-electrons, $t_f^0$. Rather it is a generic emergent term generated by fluctuations. Theoretically, it can be straightforwardly obtained by decoupling the RKKY interactions explicitly in $SU(N)$ mean-field theory [44, 45], as in $U(1)$ spin liquids. [1] Excitations are confined to the Hilbert space in Fig. 1(a) by the constraint, $\sum_\sigma b_{j\sigma}^\dagger b_{j\sigma} + \sum_\alpha f_{j\alpha}^\dagger f_{j\alpha} = 1$.

A key consequence of this auxiliary particle representation is an emergent $U(1)$ gauge symmetry:

$$b_{j\sigma} \to b_{j\sigma} \mathrm{e}^{-i\xi_j}, \; f_{j\alpha} \to f_{j\alpha} \mathrm{e}^{-i\xi_j}, \; t_{f,ij} \to t_{f,ij} \mathrm{e}^{i(\xi_i - \xi_j)}. \tag{2}$$

In the large-$N$ limit, there are two non-gauge-invariant order parameters, $b_{j\sigma}$ and $t_{f,ij}$. For simplicity, we always chose a uniform $t_{f,ij}$ that breaks no symmetries on its own. The hastatic order parameter is naively an $SU(2)$ spinor,

$$\mathbf{b}_j = |b_j| \mathrm{e}^{i\chi_j} \begin{pmatrix} \cos\frac{\theta_j}{2} \mathrm{e}^{i\phi_j/2} \\ \sin\frac{\theta_j}{2} \mathrm{e}^{-i\phi_j/2} \end{pmatrix}. \tag{3}$$

As the spinor is not gauge invariant, its spinorial nature is washed out by gauge fluctuations for any finite $N$, which also affects the gauge dependent $t_f$. As real symmetries are additionally broken, there are real order parameters, but these must be gauge invariant combinations of the original gauge dependent quantities.

## 3 Order parameters

The broken symmetries are captured by gauge invariant order parameters bilinear in $\mathbf{b}_j$. These are vectorial, but carry unmistakable fingerprints of their spinorial origin. As these break different symmetries, they may develop in stages beyond mean-field theory [see Fig. 2(a)].

- $n_{b,i} = \langle \mathbf{b}_i^\dagger \mathbf{b}_i \rangle$ is the on-site excited state occupation, which breaks no symmetries and mimics the single-channel Kondo effect, reproducing the heavy Fermi liquid signatures above $T_{HO}$ [1, 7, 34].

- $\vec{\Psi}_i = \langle \mathbf{b}_i^\dagger \vec{\sigma} \mathbf{b}_i \rangle$ are the on-site moments of the excited doublet, which correspond to the $SO(3)$ composite order parameter of the two-channel Kondo model [46–49]. The arrangement of $\vec{\Psi}_i$ may break spatial symmetries, leading to ferrohastatic (FH) or antiferrohastatic (AFH) order. In tetragonal symmetry, $\vec{\Psi}$ is reducible, with $\vec{\Psi} = \Psi_z \oplus \vec{\Psi}_\perp = m\Gamma_2^+ \oplus m\Gamma_5^+$. $m\Gamma_i^+$ labels the different irreducible representations (irreps) that preserve inversion ($+$), translation ($\Gamma$) and break time-reversal ($m$); here these are the dipolar moments, $\langle J_z \rangle$ ($m\Gamma_2^+$) and $\langle \vec{J}_\perp \rangle$ ($m\Gamma_5^+$).

---

[1]Alternately $t_f$ may be found by treating the constrained hopping of the original f-electrons, which pick up additional auxiliary boson correlations, $t_{f,ij} \sim t_f^0 \langle b_i^\dagger b_j \rangle$.

- $\vec{\Phi}_{ij} = t_{f,ij}\langle \mathbf{b}_i^\dagger \vec{\sigma} \mathbf{b}_j \rangle$ captures additional symmetries broken by the arrangement of the spinors, which generate interference between different sites, as shown in the diagram in region III of Fig. 1(c). This interference is described by the complex $\vec{\Phi}$, whose real and imaginary parts have different symmetries. $\vec{\Psi}$ is contained in $\mathrm{Re}\vec{\Phi} \otimes \mathrm{Im}\vec{\Phi}$, allowing a third order coupling. $\vec{\Phi}$ is absent for FH order, but is generic for AFH order. $t_f$ is required for gauge invariance, and consequentially $\vec{\Phi}$ is smaller than $\vec{\Psi}$ by $t_f/D$, where $D$ is the conduction electron bandwidth. Note that the emergent $t_f$ is proportional to $T_{HO}$.

Note that the original hastatic proposal [31] has only $\vec{\Psi}$, and does not actually break double-time-reversal symmetry. The original hybridization structure had four sublattices, naively breaking inversion symmetry. The additional signs were absorbed into the $f$-fermions via a gauge transformation, and the transformed $f$s were assumed to have a simple $t_f$ hopping, which implies that the original, untransformed $f$s had an inversion symmetry breaking $t_f$ hopping. Essentially, the symmetry breaking of the hybridization and the f-hopping canceled to give a singlet $\Phi$, while $\vec{\Psi}$ remained. The resulting dispersion was doubly degenerate, reflecting the double-time-reversal symmetry. In principle, this phase is always possible, but there is no a priori reason to expect it to be favored over the spinorial phases with both $\vec{\Psi}$ and $\vec{\Phi}$. Its Landau-Ginzburg physics is fully captured by $\vec{\Psi}$.

## 4   Broken symmetries and moments

Now we discuss these order parameters in the context of URu$_2$Si$_2$. The hidden order is thought to have the same wave-vector as the antiferromagnetism found under pressure, as the Fermi surfaces found in de Haas-van Alphen change very little. This order is staggered between the planes, with $\mathbf{Q} = Z = [001]$. We similarly stagger $\vec{\Psi}$, with $\vec{\Psi} = mZ_2^+ \oplus mZ_5^+$, which indicates that these moments are staggered by $\mathbf{Q}$. This $\vec{\Psi}$ order corresponds to multiple spinor arrangements differentiated by $\vec{\Phi}$, see Fig. 1(b) for two examples. Here, we consider two examples that are also uniform in the plane, but with different $\hat{c}$ axis behaviors. First we consider the naive two sublattice (2SL) case, with the spinors on the two sublattices defined as $\mathbf{b}_A$ and $\mathbf{b}_B = \theta \mathbf{b}_A$, where $\theta = i\sigma_2 K$ is the time-reversal operator, with $K$ indicating complex conjugation. As these are spinors, $\theta^2 \mathbf{b}_A = -\mathbf{b}_A$, this order is not invariant under time-reversal followed by translation (or any symmetry operation). To preserve a time-reversal-like symmetry, *four* sublattices (4SL) are required, with $\mathbf{b}_C = \theta^2 \mathbf{b}_A = -\mathbf{b}_A$ and $\mathbf{b}_D = \theta^3 \mathbf{b}_A = -\mathbf{b}_B$. Both 2SL and 4SL orders are plausible, with one breaking time-reversal uniformly and the other breaking inversion. Analogues of both appear in cubic hastatic order [50]. $\vec{\Psi}$ and $\vec{\Phi}$ are then,

$$\vec{\Psi} = \langle \mathbf{b}_A^\dagger \vec{\sigma} \mathbf{b}_A \rangle - \langle \mathbf{b}_B^\dagger \vec{\sigma} \mathbf{b}_B \rangle, \quad \vec{\Phi} = t_f \langle \mathbf{b}_A^\dagger \vec{\sigma} \mathbf{b}_B \rangle, \tag{4}$$

with corresponding momentum space versions presented in Appendix A.

We identify $\vec{\Phi}_{2SL} = (m\Gamma_2^+ \oplus m\Gamma_5^+) \oplus Z_5^+$, and $\vec{\Phi}_{4SL} = (mZ_2^- \oplus mZ_5^-) \oplus \Gamma_5^-$, where the first (second) terms are the real (imaginary) components. Again, $\Gamma$ and $Z$ indicate uniform or c-axis staggered moments, $m$ indicates moments that break time-reversal, and $\pm$ indicates moments even or odd under inversion symmetry.

Table 1 gives the four possible order parameters and associated moments in region IV (where both $\vec{\Psi}$ and $\vec{\Phi}$ are nonzero); regions II and III have subsets of these, as $\vec{\Phi}$ or $\vec{\Psi}$ are zero, respectively.

As all region IV phases have an in-plane dipolar component, tetragonal symmetry breaking is ubiquitous when both $\vec{\Psi}$ and $\vec{\Phi}$ are nonzero, even for $\vec{\Psi}$ along $\hat{c}$. These moments

|  | $\Psi_z = mZ_2^+$ <br> staggered $m_z$ | $\vec{\Psi}_\perp = mZ_5^+$ <br> staggered $\vec{m}_\perp$ |
|---|---|---|
| 2SL [$\mathrm{Im}\vec{\Phi} = Z_5^+$] <br> staggered $(Q_{xz}, Q_{yz})$ | $\mathrm{Re}\vec{\Phi} = m\Gamma_2^+$ <br> uniform $\vec{m}_\perp$ | $\mathrm{Re}\vec{\Phi} = m\Gamma_5^+$ <br> uniform $m_z$ |
| 4SL [$\mathrm{Im}\vec{\Phi} = \Gamma_5^-$] <br> uniform $\vec{p}_\perp$ | $\mathrm{Re}\vec{\Phi} = mZ_5^-$ <br> staggered $\vec{\Omega}_\perp$ | $\mathrm{Re}\vec{\Phi} = mZ_2^-$ <br> staggered $\Omega_z$ |

Table 1: Possible AFH phases and associated moments in region IV of Fig. 1. In region II, only $\vec{\Psi}$ moments are present, while in region III only $\mathrm{Re}\vec{\Phi}$ or $\mathrm{Im}\vec{\Phi}$ moments are nonzero. $m, p, \Omega$ refer to magnetic, electric and toroidal dipoles, respectively, while $Q$ indicates electric quadrupoles. In-plane moments are susceptible to being washed out by disorder. The full set of primary and secondary order parameters is given in Appendix C.

are all susceptible to being washed out by disorder, and so we highlight the robust consequences of $\vec{\Phi}$ in the $\vec{\Psi}_\perp$ phase, where uniform magnetic dipoles, $m_z$ or staggered toroidal dipoles, $\Omega_z$ are expected in 2SL and 4SL, respectively. Microscopic theory suggests that these moments are tiny, $\sim T_{HO} t_f / D^2$. Fortunately, Kerr effect or second harmonic generation measurements should be sufficiently sensitive, and Kerr measurements tantalizingly suggest that $m_z$ develops slightly above $T_{HO}$ [12].

# 5 Antiferrohastatic Landau theory

Having understood the symmetries, we now turn to the Landau theory to discuss the thermodynamic and other responses. As $t_f/D$ suppresses the effects of $\vec{\Phi}$, we focus on the $\vec{\Psi}$ transition from region I to II. The additional transition to region IV is also second order, but is practically undetectable, as the specific heat jump is suppressed by $\sim (t_f/D)^2$ compared to the first jump, and is expected to be within the experimental noise, $\lesssim 3\mathrm{mJ\,mol}^{-1}\mathrm{K}^{-1}$ (see Appendix D). We therefore consider,

$$F_\Psi = \alpha_\perp (T - T_c^\perp)|\vec{\Psi}_\perp|^2 + \alpha_z(T - T_c^z)\Psi_z^2 + u_\perp|\vec{\Psi}_\perp|^4 + u_z\Psi_z^4 - v_1(\Psi_{\Gamma_4}^2)^2 + v_2\Psi_z^2|\vec{\Psi}_\perp|^2. \quad (5)$$

$F_\Psi$ describes two independent order parameters, $\Psi_z$ and $\vec{\Psi}_\perp$ that couple quadratically, and $\Psi_{\Gamma_4}^2 = 2\Psi_x\Psi_y$.

The parameter choices are guided by the microscopic calculations [33, 41], as discussed in Appendix E. We know that $T_c^\perp = T_c^z \equiv T_{HO}$ and that the order parameters repel ($v_2 \gg 1$). A first order transition between $\Psi_z$ and $\vec{\Psi}_\perp$ can be induced by varying $u_z - u_\perp = u(p - p_c^0)$; pressure tunes $V_6/V_7$, which induces just such a transition microscopically. The sign of $v_1$ determines the pinning of $\phi$ and nature of the tetragonal symmetry breaking.

As tetragonal symmetry is generically broken, we consider secondary ferroquadrupolar order parameters, $R_{\Gamma_3}$, $R_{\Gamma_4}$ and $\vec{R}_{\Gamma_5}$ that couple linearly to strain as shown in Appendix E:

$$\epsilon_{x^2-y^2} = \frac{g_3 R_{\Gamma_3}}{c_{11} - c_{12}}, \; \epsilon_{xy} = \frac{g_4 R_{\Gamma_4}}{2c_{66}}, \; (\epsilon_{xz}, \epsilon_{yz}) = \frac{g_5 \vec{R}_{\Gamma_5}}{2c_{44}}. \quad (6)$$

$R_{\Gamma_3}$ and $R_{\Gamma_4}$ occur naturally in the $\Gamma_5$ doublet, and couple to bilinears of $\vec{\Psi}_\perp$: $\Psi_{\Gamma_3}^2 = \Psi_x^2 - \Psi_y^2$ and $\Psi_{\Gamma_4}^2 = 2\Psi_x\Psi_y$, while $\vec{R}_{\Gamma_5}$ shear strain couples to $\Psi_z\vec{\Psi}_\perp$. These give,

$$F_R = \sum_{i=3,4,5} \left( \alpha_R^{(i)} R_{\Gamma_i}^2 + u_R^{(i)} R_{\Gamma_i}^4 \right), \quad F_{\Psi-R} = \gamma_3 R_{\Gamma_3}\Psi_{\Gamma_3}^2 + \gamma_4 R_{\Gamma_4}\Psi_{\Gamma_4}^2 + \gamma_5 \Psi_z\vec{\Psi}_\perp \times \vec{R}_{\Gamma_5}. \quad (7)$$

Finally, we consider the coupling to magnetic field, $\vec{h}$, which has $m\Gamma_2^+ \oplus m\Gamma_5^+$ symmetry, like FH order. Bilinears of $\vec{h}_\perp$ also break tetragonal symmetry: $h_{\Gamma_3}^2 = h_x^2 - h_y^2$ and $h_{\Gamma_4}^2 = 2h_x h_y$. The additional free energy terms are,

$$F_{\Psi\text{-}h} = u_h^{(1)} h_z^2 \Psi_z^2 + u_h^{(2)} |\vec{h}_\perp|^2 \Psi_z^2 + u_h^{(3)} h_z^2 |\vec{\Psi}_\perp|^2 + u_h^{(4)} |\vec{h}_\perp|^2 |\vec{\Psi}_\perp|^2 + v_h^{(3)} h_{\Gamma_3}^2 \Psi_{\Gamma_3}^2 + v_h^{(4)} h_{\Gamma_4}^2 \Psi_{\Gamma_4}^2. \quad (8)$$

The ferroquadrupolar orders also couple to $\vec{h}$ via $F_{R-h}$, as given in the Appendix E.

The physics of AFH order can be explored by fixing a set of parameters and minimizing the free energy to obtain temperature, pressure and transverse/longitudinal field phase diagrams, as well as thermodynamic responses across the various transitions (details in Appendix E). These quantities compare favorably to the experimental literature on $URu_2Si_2$, where the HO may be identified with $AFH_\perp$ ($\vec{\Psi}_\perp$) order, and the antiferromagnet (AFM) identified with $AFH_z$ ($\Psi_z$) order. Remember, symmetry-wise, $\vec{\Psi}$ and AFM orders are identical, but the microscopic origins give vastly different parameter sets. Notably, the calculated U $5f^2$ moments in the $AFH_z$ phase are $\sim .5\mu_B$, consistent with neutron measurements [51, 52], while the AFH $\vec{\Psi}_\perp$ phase has much smaller in-plane U $5f^3$ moments $\lesssim .01\mu_B$ for a fairly substantial mixed valency ($\sim 20\%$) [31, 41].

## 5.1 Tetragonal symmetry breaking in $URu_2Si_2$.

There are two distinct in-plane AFH orders, $AFH_{x^2-y^2}$ and $AFH_{xy}$, with $\Psi_{\Gamma_3}^2$ $\Psi_{\Gamma_4}^2$ nonzero, respectively. This tetragonal symmetry breaking actually has three experimental manifestations with different Landau coefficients: the coupling to the lattice via ferroquadrupolar order parameters ($\gamma_{3,4}$); the coupling to magnetic field ($v_h^{(3,4)}$) that yields anisotropic magnetic susceptibilities; and the coupling to an electronic nematic order parameter affecting the resistivity ($\zeta_{3,4}$, not shown; see Appendix G). The connection to microscopics is particularly useful here, as the induced electric quadrupolar moments, $R_{3,4} = \gamma_{3,4}\Psi_{\Gamma_{3,4}}^2$ are tiny, even when $\Psi_{\Gamma_{3,4}}^2$ is relatively large, suggesting that the coupling constants are of order $(T_{HO}/D)^2 \sim .001$. The anisotropic field couplings, $v_h^{(3,4)}$ are similarly suppressed [33], although typically found to be an order of magnitude larger microscopically [41]. At the same time, the tetragonal symmetry breaking near the Fermi surface is of order one, which would give substantial resistivity anisotropy signatures. As we discuss below, the tetragonal symmetry breaking may be washed out by disorder, but the signals at and above the transition remain, and tetragonal symmetry breaking may be restored by applied field or strain.

The coupling to the lattice ($\gamma_{3,4}$) manifests both as tiny jumps in the relevant elastic coefficients that have not been seen [18] (see Appendix F for details), and tiny directional jumps in the thermal expansion or magnetostriction (see Appendix E); higher order terms like $R_{\Gamma_i}^2|\Psi|^2$, neglected here are not necessarily small and give kinks in all elastic coefficients [18]. Below the transition, the small ferroquadrupolar moments lead to a orthorhombic distortion. The experimental evidence here is mixed. One x-ray experiment [15] has observed a $\Gamma_4$ ($=B_{2g} = Fmmm$ space group) distortion, but other results do not find a distortion in the HO region, but instead find the opposite ($\Gamma_3 = B_{1g} = Immm$) distortion at higher pressures, with $T_R \gg T_{HO}$ [17, 40] developing rapidly near the critical pressure between the HO and AFM. As an aside, we can straightforwardly consider this additional orthorhombicity, shown in Fig. 2(a) by giving $R_{\Gamma_3}$ a transition temperature $T_{R3}(p)$ that vanishes at $p = p_R$. This independent orthorhombic transition leads to additional transitions within the $\vec{\Psi}_\perp$ phase where $\phi$ changes; specific heat and structural signatures are weak due to the near constant $|\vec{\Psi}_\perp|$ and small $\gamma_{3,4}$, respectively, but the transition would be visible in elastoresistivity. The close coincidence of orthorhombic and AFM transitions in pressure would be accidental in this scenario.

Unlike the small coupling to the lattice, the large coupling to electronic nematicity gives significant jumps in the elastoresistivity at $T_{HO}$, in $m_{11} - m_{12}$ ($\Psi_{\Gamma_3}^2$), or $m_{66}$ ($\Psi_{\Gamma_4}^2$) [16,19] (see Appendix G); and $\Gamma_4$ nematicity captures the cyclotron mass anisotropy with heavier masses along [110] [53].

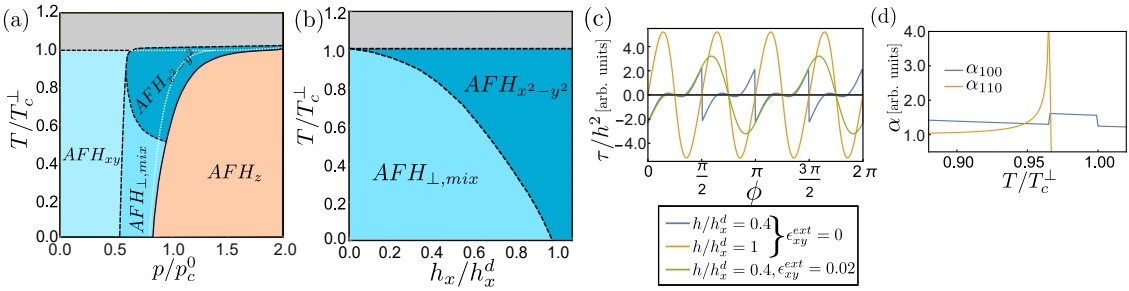

Figure 2: Transitions in the in-plane moment direction due to field or independent orthorhombic order parameters. (a) $T - p$ phase diagram including the onset of independent orthorhombicity ($R_{\Gamma_3}$) at $p \approx 0.5$; white dashed lines correspond to the original transitions. (b) $T - h_x$ phase diagram at $p = 0$, showing the field-locking transition. (c) Magnetic torque curves for different in-plane fields at $T/T_c^\perp = 0.4$, with and without small external $\epsilon_{\Gamma_4}$ strain. (d) Thermal expansion coefficients are sensitive to both transitions, at main hastatic transition ($T_{HO}$) and the field-locking transition for $h_x/h_x^d = 0.1$ ($T_c' \approx 0.96 T_{HO}$). Parameters for all plots are given in Appendix E.

The coupling to magnetic field allows the magnetic susceptibility to break tetragonal symmetry, with either $\chi_{xx} - \chi_{yy}$ or $\chi_{xy}$ nonzero for $\Psi_{\Gamma_{3,4}}^2$, respectively, as found in torque magnetometry measurements [14]. The magnetic susceptibility [14] and elastoresistivity [16] data suggest that $\Gamma_4$ symmetry breaking is favored by $v_1 > 0$.

## 5.2 Finite field behavior: field locking transition and torque magnetometry

To properly understand the torque magnetometry, we consider finite transverse magnetic fields in the absence of disorder. As the pinning is weak, it is possible to reorient $\vec{\Psi}_\perp$ via the quadratic coupling $v_h^{(3,4)}$. If the zero-field order is $\Gamma_4$ (AFH$_{xy}$), $\vec{h} = h_x \hat{x}$ favors the "field-locked" $\Gamma_3$ (AFH$_{x^2-y^2}$) at temperatures and fields beyond $T_c'(h_x)$. Above $T_c'$, only $\Psi_{\Gamma_3}^2$ is present and $\Psi_{\Gamma_4}^2$ turns on via a second order transition, as shown in Fig. 2 (b).

To estimate the field-locking transition temperature and model response functions to lowest order in $h_x$, we use the simplified free energy,

$$F_m = \alpha_\perp (T - T_c^\perp)|\vec{\Psi}_\perp|^2 + u_\perp |\vec{\Psi}_\perp|^4 - v_1 (\Psi_{\Gamma_4}^2)^2 - v_h^{(3)} h_{\Gamma_3}^2 \Psi_{\Gamma_3}^2. \tag{9}$$

We can minimize the free energy in both the high temperature field-locked phase, where the order parameter is:

$$\Psi_x^{(0)}|_{x^2-y^2} = \sqrt{-\frac{\alpha_\perp(T - T_c^\perp) - v_h^{(3)} h_x^2}{2u_\perp}}, \; \Psi_y^{(0)}|_{x^2-y^2} = 0. \tag{10}$$

In the lower temperature mixed phase, we acquire an additional component,

$$\Psi_x^{(0)}|_{\perp,mix} = \sqrt{-\frac{\alpha_\perp(T-T_c^\perp) - v_h^{(3)}h_x^2\left(\frac{u_\perp}{v_1}-1\right)}{4u_\perp - v_1}},$$

$$\Psi_y^{(0)}|_{\perp,mix} = \sqrt{-\frac{\alpha_\perp(T-T_c^\perp) + v_h^{(3)}h_x^2\left(\frac{u_\perp}{v_1}-1\right)}{4u_\perp - v_1}}. \tag{11}$$

The second order transition occurs when $\Psi_y^{(0)}$ becomes nonzero, at

$$\frac{T_c'}{T_c^\perp} = 1 - \frac{v_h^{(3)}h_x^2}{\alpha_\perp T_c^\perp}\left(1 + \frac{4u_\perp}{|v_1|}\right) \approx 1 - \left(\frac{h_x}{h_z^c}\right)^2. \tag{12}$$

We estimate the $\mathcal{O}(1)$ prefactor of the field dependent term from the microscopics, as $4u_\perp/|v_1| \sim (D/T_{HO})^2$, while $v_h^{(3)}/(\alpha_\perp T_c^\perp) \sim v_h^{(3)}/[u_h^{(3)}(h_z^c)^2] \sim (T_{HO}/D)^2$. This critical field is thus proportional to the c-axis critical field, $h_z^c \approx 35$T [54], and $h_x \geq 10$T is likely required to distinguish the two transitions.

In fact, the overall magnitude of $|\Psi|$ across the phase transition is almost perfectly smooth, despite large changes in $\Psi_{x,y}$. Standard bulk probes, like specific heat, are therefore unlikely to detect this transition. Indeed, the specific heat jumps at two transitions are given by,

$$\Delta C|_{T_{HO}} = \left(-T\frac{\partial^2 F_m|_{x^2-y^2}}{\partial T^2}\right)_{T_c}, \quad \Delta C|_{T_c'} = \left(-T\frac{\partial^2\left(F_m|_{\perp,mix} - F_m|_{x^2-y^2}\right)}{\partial T^2}\right)_{T_c'}, \tag{13}$$

which indicates that the ratio of the two,

$$\frac{\Delta C|_{T_c'}}{\Delta C|_{T_c}} = \frac{v_1}{4u_\perp - v_1} \approx \frac{v_1}{4u_\perp} \approx \left(\frac{T_{HO}}{D}\right)^2, \tag{14}$$

is very small, and $\Delta C|_{T_c'}/T_c' \lesssim 0.2$mJ mol$^{-1}$ K$^{-2}$ will be within experimental noise [55,56] and thus undetectable.

While standard bulk probes will be blind to this transition, properties sensitive to either $\Psi_{\Gamma_3}^2$ or $\Psi_{\Gamma_4}^2$, like elastoresistivity will have larger jumps at both transitions. There will be a jump in $m_{11} - m_{12}$ at $T_{HO}$ and in both $m_{11} - m_{12}$ and $m_{66}$ at $T_c'$. These behave similarly to the thermal expansion, although the thermal expansion itself is likely difficult to detect due to the smallness of ferroquadrupolar couplings ($\gamma_{3,4} \sim (T_{HO}/D)^2$) and the possibility of multi-domain cancellation below $T_{HO}$.

To treat these directionally dependent quantities analytically, we include the ferro-quadrupolar terms,

$$\delta F_m = \alpha_R^{(3)}R_{\Gamma_3}^2 + \alpha_R^{(4)}R_{\Gamma_4}^2 + \gamma_3 R_{\Gamma_3}\Psi_{\Gamma_3}^2 + \gamma_4 R_{\Gamma_4}\Psi_{\Gamma_4}^2, \tag{15}$$

and calculate the thermal expansion coefficients according to Appendix E to leading order in $\gamma_3$ and $\gamma_4$. Using Eq. (10)-(12), we find jumps of the same magnitude (but opposite sign) in $\alpha_{100}$ at both the main transition and at the field-locking one:

$$\Delta\alpha_{100}|_{T_{HO},T_c'} \approx \frac{\gamma_3}{\alpha_R^{(3)}}\left(\frac{\partial\Psi_{\Gamma_3}^2}{\partial T}\right)_{T_{HO},T_c'} \approx \pm\frac{\gamma_3}{\alpha_R^{(3)}}\frac{\alpha_\perp}{2u}, \tag{16}$$

while $\alpha_{110}$ shows a $(T_c' - T)^{-1/2}$ divergence only below the field-locking transition,

$$\Delta\alpha_{110}|_{T_{HO}} = 0, \qquad \alpha_{110}|_{T \to T_c'+} = 0,$$

$$\alpha_{110}|_{T \to T_c'-} \approx \frac{\gamma_4}{\alpha_R^{(4)}} \left(\frac{\partial\Psi_{\Gamma_4}^2}{\partial T}\right)_{T \to T_c'-} \approx \frac{\gamma_4}{\alpha_R^{(4)}} \frac{\alpha_\perp}{2\sqrt{2}u} \sqrt{\frac{T_c^\perp - T_c'}{T_c' - T}}. \qquad (17)$$

These are calculated numerically for the full Landau free energy and shown in Fig. 2(d).

The pinning also affects the torque magnetometry at lower fields, which is used to measure the magnetic susceptibility matrix elements. We numerically simulate torque curves with our full $\vec{\Psi}$ free energy using:

$$\vec{\tau} = \vec{M} \times \vec{h} = -\frac{\partial F}{\partial\vec{h}} \times \vec{h}, \qquad (18)$$

with results shown in Fig. 2(c). For a system with pure $\mathbb{Z}_4$ pinning, hysteretic behavior is found for $h_x < h_x^d(T)$. For larger fields, above the field-locking critical field there is no longer a two-fold component of the torque and the torque curves are entirely four-fold in character as the order parameter "follows" the field. Experiments, however, have shown a substantial, but smooth two-fold anisotropic response, although only in small crystals, where it has been attributed to the $\mathbb{Z}_2$ pinning effects of surface strains. We include these strains as an external $\Gamma_4$ strain ($\epsilon_{xy}^{ext}$) that couples as $\delta F_\epsilon^{ext} = -\lambda_{xy}\epsilon_{xy}^{ext}\Psi_{\Gamma_4}^2$ and replaces the hysteresis by a two-fold anisotropic response similar to the experiment.

# 6 Fluctuations and disorder

While the weak pinning already affects the mean-field physics, it is even more important when discussing fluctuations. The in-plane moments are governed by a 3D XY model with weak $\mathbb{Z}_4$ pinning [57], where we estimate the pinning for $URu_2Si_2$, $v_1/u_\perp \sim (T_{HO}/D)^2 \sim$ .001 for $T_{HO}/D \sim 1/30$ [31, 41]. The 3D XY model is well known to be disordered by infinitesimal random fields that couple linearly to the massless transverse fluctuations [58, 59], and here we will show that the AFH $\vec{\Psi}_\perp$ does couple linearly to random strains: the fluctuations of $\Gamma_4$ order (AFH$_{xy}$) couple to uniform $\Gamma_3$ strains, and vice versa for $\Gamma_3$ (AFH$_{x^2-y^2}$) order. Thus, we expect the in-plane order to be lost if disorder is stronger than the pinning [60–62]. It is important to note that hastatic order itself survives, even in the XY limit, as the moments are not pre-formed, like in the pure XY spin model. Instead, the hastatic moment magnitude develops ($\langle b^\dagger b\rangle$) and choose to lie in the XY plane ($|\vec{\Psi}_\perp|$) at $T_c$, with large barriers for out of plane fluctuations. Even the barriers for translation symmetry breaking are large, and so the AFH nature is also robust. Therefore, most signatures of hastatic order will remain, with only the tetragonal symmetry breaking washed out by disorder.

We can examine this process within Landau-Ginzburg theory, where we will focus on only the $\vec{\Psi}$ terms,

$$\mathcal{L} = c_\perp \left|\nabla\vec{\Psi}_\perp\right|^2 + c_z(\nabla\Psi_z)^2 + r_\perp|\vec{\Psi}_\perp|^2 + r_z\Psi_z^2 - \sum_{i=3,4}\left(\lambda_i\epsilon_{\Gamma_i} + v_h^{(i)}h_{\Gamma_i}^2\right)\vec{\Psi}_{\Gamma_i}^2$$

$$+ u_\perp|\vec{\Psi}_\perp|^4 + u_z\Psi_z^4 - v_1(\Psi_{\Gamma_4}^2)^2 + v_2\Psi_z^2|\vec{\Psi}_\perp|^2, \qquad (19)$$

where $r_\perp = \alpha_\perp(T - T_c^\perp)$ and $r_z = \alpha_z(T - T_c^z)$. We expand around the $AFH_{xy}$ ground state, initially with no external strain, $\epsilon_{\Gamma_i}$ or field, $\vec{h}$. We write $\vec{\Psi} = (\Psi_0 + \delta\Psi_l + \delta\Psi_t, \Psi_0 +$

$\delta\Psi_l - \delta\Psi_t, \delta\Psi_z)$, with $\Psi_0 = \sqrt{-\frac{r_\perp}{4u_\perp - v_1}}$. Here, we have decomposed the fluctuations into longitudinal ($\delta\Psi_l$), in-plane transverse ($\delta\Psi_t$) and out of plane transverse ($\delta\Psi_z$). Note that expanding around the $AFH_{x^2-y^2}$ state gives similar results, where the role of $\Gamma_3$ and $\Gamma_4$ strains will be swapped. We now expand to second order in the fluctuation fields, using

$$
\begin{aligned}
\left|\vec{\Psi}_\perp\right|^2 &\approx 2\Psi_0^2 + 2(\delta\Psi_l)^2 + 2(\delta\Psi_t)^2 + 4\Psi_0\delta\Psi_l, \\
\left|\vec{\Psi}_\perp\right|^4 &\approx 4\Psi_0^4 + 24\Psi_0^2(\delta\Psi_l)^2 + 16\Psi_0^3\delta\Psi_l + 8\Psi_0^2(\delta\Psi_t)^2, \\
(\Psi_{\Gamma_4}^2)^2 &\approx \Psi_0^4 + 6\Psi_0^2(\delta\Psi_l)^2 - 2\Psi_0^2(\delta\Psi_t)^2 + 4\Psi_0^3\delta\Psi_l.
\end{aligned}
\tag{20}
$$

The three fluctuation fields completely decouple, $\mathcal{L} = \mathcal{L}[\delta\Psi_z] + \mathcal{L}[\delta\Psi_l] + \mathcal{L}[\delta\Psi_t]$, with

$$
\begin{aligned}
\mathcal{L}[\delta\Psi_z] &= c_z\left(\nabla\delta\Psi_z\right)^2 + \left(r_z - 2v_2\frac{r_\perp}{4u_\perp - v_1}\right)(\delta\Psi_z)^2, \\
\mathcal{L}[\delta\Psi_l] &= c_\perp(\nabla\delta\Psi_l)^2 - 2r_\perp(\delta\Psi_l)^2, \\
\mathcal{L}[\delta\Psi_t] &= c_\perp(\nabla\delta\Psi_t)^2 - \frac{r_\perp v_1}{4u_\perp - v_1}(\delta\Psi_t)^2.
\end{aligned}
\tag{21}
$$

The masses of the fluctuation fields can be read off directly, using that an action with $c(\nabla\phi)^2 + \beta\phi^2$ corresponds to scalar field of mass $m[\phi] = \sqrt{\beta/c}$. We find two "heavy" fluctuation fields with comparable masses ($\delta\Psi_z$ and $\delta\Psi_l$) and one "light" field ($\delta\Psi_t$), with mass ratios,

$$
\begin{aligned}
\frac{m_z}{m_l} &= \sqrt{\frac{\left|r_z - 2v_2\frac{r_\perp}{4u_\perp - v_1}\right|}{|2r_\perp|}\frac{c_\perp}{c_z}} \approx \sqrt{\left(\frac{2v_2}{4u_\perp - |v_1|} - \frac{1}{2}\right)\frac{c_\perp}{c_z}} \sim \mathcal{O}(1), \\
\frac{m_t}{m_l} &= \sqrt{\frac{|v_1|}{4u_\perp - |v_1|}} \approx \sqrt{\frac{|v_1|}{4u_\perp}} \sim \mathcal{O}\left(\frac{T_{HO}}{D}\right).
\end{aligned}
\tag{22}
$$

These fluctuations have large coherence lengths, $\xi_t/\xi_l = m_l/m_t \sim \mathcal{O}(D/T_{HO}) \sim 100$ unit cells. [2]. Note that a similar analysis can also be done in the $AFH_z$ phase including $\vec{\Phi}$, where $\vec{\Phi}$ breaks tetragonal symmetry and has similarly light transverse in-plane fluctuations.

Next we consider the coupling of these light transverse fluctuations to strain,

$$
\mathcal{L}_{\Psi-\epsilon} = \lambda_3\epsilon_{\Gamma_3}\Psi_{\Gamma_3}^2 + \lambda_4\epsilon_{\Gamma_4}\Psi_{\Gamma_4}^2.
\tag{23}
$$

Expanding in terms of light fluctuations using $\Psi_{\Gamma_3}^2 \approx 4\Psi_0\delta\Psi_t$ and Eq. (20) for $\Psi_{\Gamma_4}^2$, we find that the light transverse fluctuations couple linearly to $\Gamma_3$ strain, as

$$
\mathcal{L}_{\delta\Psi_t-\epsilon_{\Gamma_3}} = 4\lambda_3\Psi_0\epsilon_{\Gamma_3}\delta\Psi_t,
\tag{24}
$$

to leading order in $\delta\Psi_t$. Thus, for [110] pinning, we find that the light transverse fluctuations couple linearly to random $\epsilon_{\Gamma_3}$ strain, but quadratically to random $\epsilon_{\Gamma_4}$ strain; the situation is reversed for [100] pinning.

Following the original argument of Imry and Ma [58], we consider how random strains can disorder the in-plane order. While the overall mean $\langle\epsilon_{\Gamma_3}\rangle = 0$, the local average over any finite region of volume $L^d$ is nonzero and scales as $\langle\Delta\epsilon_{\Gamma_3}^2\rangle \sim \alpha^2 L^d$, where $\alpha$ parameterizes the disorder strength. This finite region can therefore gain an energy $aL^{d/2}$ (with $a \sim \lambda_3\Psi_0\alpha$) by forming a domain where $\delta\Psi_t$ aligns with the average local random

---

[2]We expect $T_{HO}/D$ to take values in range 0.01-0.03 as it determines the degree of mixed valence.

field. Domain walls of length $L$ cost an energy $cL^{d-1}$, where $c \sim m_t^2$, giving an overall domain cost, $\Delta E_d(L) = -aL^{3/2} + cL^2$, where we set $d = 3$. As long as disorder is sufficiently weak, for $L < L_0 \sim (a/c)^2$, the transverse fluctuations, $\delta\Psi_t$ align with the local average strain and order is lost on these shorter length scales, where $L_0$ acts as a new, larger coherence length. For stronger disorder, the full random field four-state clock model should be treated [61]; we simply assume that the tetragonal ordering temperature, $T_I(\alpha)$ decreases monotonically from $T_{HO}$ for $\alpha = 0$ to zero at $\alpha_c$, where $\alpha_c \sim c \sim m_t^2$. Therefore, as external $\Gamma_4$ or $h_{[110]}$ are applied, $m_t$ increases and $T_I(\vec{h}_\perp)$ can rise from zero beyond a critical field to eventually meet $T_{HO}$, as shown in Fig. 3. Note that the field-locking transition previously discussed is completely washed out if the zero-field ground state is disordered, as the smaller barrier does not increase with field.

If samples are sufficiently disordered, $\alpha > \alpha_c$ the in-plane moments will be disordered, consistent with neutron measurements [37–39] and the absence of $\chi_{xy}$ and similar signals in larger samples [14]. Moreover, the remaining random local fields would be consistent with $\mu$SR [63] and NMR [42].

Most interestingly, $m_t$ and thus $\alpha_c$ can be enhanced substantially by external strain ($\epsilon_i$) or transverse field ($\vec{h}_\perp$), making it possible to increase $\alpha_c > \alpha$. This increase causes a transition where $\vec{\Psi}_\perp$ itself orders, always breaking more symmetries than were applied. We label this transition with $T_I(\epsilon_i, \vec{h}_\perp)$ This transition is likely difficult to observe with non-symmetry-breaking signals, just like the field locking transition, but should be visible in torque magnetometry, elastoresistivity and neutron diffraction, although the staggered moments remain very small.

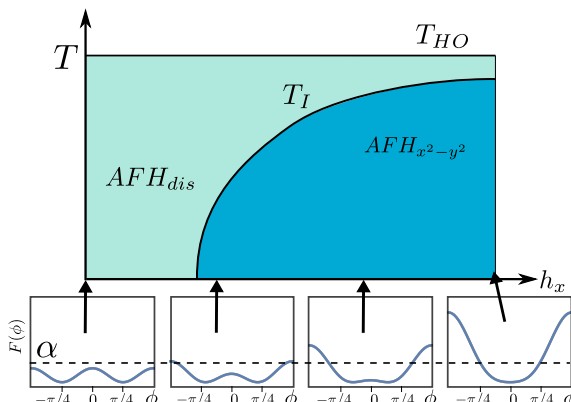

Figure 3:    (Top) A cartoon phase diagram in transverse field ($h_x$) with random strain disorder ($\alpha$) strong enough to wash out the zero-field tetragonal symmetry breaking. In-plane moments form at $T_{HO}$, where most hastatic signatures onset, but do not order until the magnetic field sufficiently increases the largest barrier between minima in $F(\phi)$, which occurs at $T_I(h_x)$. (Bottom) Sketches of the free energy dependence as a function of $\phi$ for different $h_x$, showing the evolution of the barrier heights.

# 7   Conclusion

In this paper, we showed that AFH order supports not a single, spinorial order parameter, $\langle b_{j\sigma} \rangle$ but three conventional order parameters: the scalar amplitude, $n_b$ that develops as a crossover and induces a heavy Fermi liquid; the on-site moment, $\vec{\Psi}$ that breaks symmetries like an AFM, but with significantly different Landau parameters; and intersite interference terms, $\vec{\Phi}$ that isolate the signatures of the underlying spinorial nature. We discussed how

the weak coupling of $\vec{\Psi}$ to the lattice compared to electronic quantities reconciles disparate experimental measurements of tetragonal symmetry breaking at the transition. Finally, we argued that weak pinning leads to soft transverse fluctuations that couple linearly to random strains and may disorder the in-plane moments, without totally destroying the order. We predict that the moments can be restored by stiffening the fluctuations via transverse field, with a new transition visible in elastoresistivity and neutron diffraction.

Our Landau-Ginzburg theory is consistent with previous experiments on $URu_2Si_2$, and additionally predicts:

- If $\vec{\Phi}$ orders, we expect small, but robust uniform magnetic dipoles, $m_z$ or staggered toroidal dipoles, $\Omega_z$ in the HO, for 2SL and 4SL orders, respectively; these could be seen with Kerr effect or second harmonic generation. They may onset above or below $T_{HO}$, and may be sample dependent, as the barriers between 2SL and 4SL are weak [41,50].

- We predict new transitions within the HO phase for large *transverse* fields. If the disorder is strong enough to wash out the tetragonal symmetry breaking at zero field, there will generically be an ordering temperature, $T_I$ where the applied $\mathbb{Z}_2$ pinning overcomes the Imry-Ma disordering mechanism. Alternately, if the disorder is not so strong, for applied strain, [100] perpendicular to the preferred [110] orientation, there will be a field-locking transition, $T_m$ above which the moments align with the field and below which they take an intermediate angle. Both transitions break symmetries, and give the strongest signals in elastoresistivity, not traditional thermodynamic probes.

Hastatic order is theoretically fascinating, as the spinor order breaks not only single, but double-time-reversal. Signatures of double-time-reversal symmetry breaking were absent in previous work, but are now captured by $\vec{\Phi}$. This new order parameter can also be present in cubic hastatic order, which may be relevant for $PrTi_2Al_{20}$ under pressure [50,64] or other materials. Future work may explore the interplay of double-time-reversal symmetry breaking with superconductivity and defects [49].

## Acknowledgements

We acknowledge stimulating discussions with Paul Canfield, Premala Chandra, Piers Coleman, Alan Goldman, Adam Kaminski, Bing Li, Peter Orth, Brad Ramshaw, Victor Quito, Ben Ueland and John van Dyke.

**Funding information** This work was supported by the U.S. Department of Energy, Office of Science, Basic Energy Sciences, under Award No. DE-SC0015891. R.F. thanks the Aspen Center for Physics, supported by the NSF Grant PHY-1607611, for their hospitality.

## A  Momentum space order parameters

In the main text, we derived the antiferrohastatic (AFH) order parameters in real space, but it is often convenient to derive them instead in momentum space, where the **Q** of the order can easily narrow down plausible order parameters. We begin with the real space

gauge invariant quantities, $\vec{\Psi}_i = \langle \mathbf{b}_i^\dagger \vec{\sigma} \mathbf{b}_i \rangle$ and $\vec{\Phi}_{ij} = t_{f,ij} \langle \mathbf{b}_i^\dagger \vec{\sigma} \mathbf{b}_j \rangle$ and Fourier transform:

$$\vec{\Psi}_{\mathbf{q}} = \frac{1}{N_s} \sum_{\mathbf{k}} \langle \mathbf{b}_{\mathbf{k+q}}^\dagger \vec{\sigma} \mathbf{b}_{\mathbf{k}} \rangle = \frac{1}{N_s} \sum_i \langle \mathbf{b}_i^\dagger \vec{\sigma} \mathbf{b}_i \rangle \mathrm{e}^{i\mathbf{q}\cdot\mathbf{R}_i}, \tag{25}$$

$$\vec{\Phi}_{\mathbf{q}} = \frac{1}{N_s^2} \sum_{\mathbf{k_1,k_2}} t_{f,\mathbf{k_1 k_2}} \langle \mathbf{b}_{\mathbf{k_1+q}}^\dagger \vec{\sigma} \mathbf{b}_{\mathbf{k_2}} \rangle = \frac{1}{N_s^2} \sum_{i,j} t_{f,ij} \langle \mathbf{b}_i^\dagger \vec{\sigma} \mathbf{b}_j \rangle \mathrm{e}^{i\mathbf{q}\cdot\mathbf{R}_i}, \tag{26}$$

where $\vec{R}_i$ denotes the real space lattice vectors, $N_s$ the number of sites, and we used the Fourier transforms,

$$\mathbf{b}_{\mathbf{k}} = \frac{1}{\sqrt{N_s}} \sum_i e^{i\mathbf{k}\cdot\mathbf{R}_i} \mathbf{b}_i,$$

$$t_{f,\mathbf{k_1 k_2}} = \frac{1}{N_s} \sum_{i,j} e^{i(\mathbf{k_1}\cdot\mathbf{R}_i - \mathbf{k_2}\cdot\mathbf{R}_j)} t_{f,ij}. \tag{27}$$

For simplicity, we assume that $t_{f,ij} = t_f \sum_{\boldsymbol{\eta}} \delta(\mathbf{R}_i + \boldsymbol{\eta} - \mathbf{R}_j)$ is uniform and only nonzero between nearest-neighbors indicated by $\boldsymbol{\eta}$. This makes $t_{f,\mathbf{k_1,k_2}} = t_f \sum_{\boldsymbol{\eta}} \mathrm{e}^{-i\mathbf{k_1}\cdot\boldsymbol{\eta}} \delta(\mathbf{k_1} - \mathbf{k_2})$, and

$$\vec{\Phi}_{\mathbf{q}} = \frac{t_f}{N_s} \sum_{\mathbf{k},\boldsymbol{\eta}} \langle \mathbf{b}_{\mathbf{k+q}}^\dagger \vec{\sigma} \mathbf{b}_{\mathbf{k}} \rangle \mathrm{e}^{-i\mathbf{k}\cdot\boldsymbol{\eta}}. \tag{28}$$

Now we turn to the specific case of $\mathrm{URu_2Si_2}$, where we assume that the spinor is uniform in the plane and modulated along $\hat{z}$ as shown in Fig. 1 in the main text. Knowing that $\mathbf{Q} = [001]$, we expect that only $\langle \boldsymbol{b_0} \rangle$ is nonzero for uniform, ferrohastatic (FH) order [65], while for the two sublattice (2SL) AFH order, $\langle \boldsymbol{b_0} \rangle$ and $\langle \boldsymbol{b_Q} \rangle$ are nonzero and for the four sublattice (4SL) AFH, $\langle \boldsymbol{b_{\pm Q/2}} \rangle$ are nonzero (with $\langle \boldsymbol{b_0} \rangle$ and $\langle \boldsymbol{b_Q} \rangle$ vanishing due to the preservation of time-reversal). In simplifying the expressions, we use $2\boldsymbol{Q} = \boldsymbol{0}$. It is convenient to rewrite these non-zero Fourier components in terms of the real space $\mathbf{b}_A$ and $\mathbf{b}_B = \hat{\theta} \mathbf{b}_A$ for the 2SL/4SL cases:

$$\mathbf{b_0} = \frac{1}{\sqrt{2}} \left( \mathbf{b}_A + \mathbf{b}_B \right), \qquad \mathbf{b_Q} = \frac{1}{\sqrt{2}} \left( \mathbf{b}_A - \mathbf{b}_B \right), \tag{29}$$

$$\mathbf{b}_{\frac{\mathbf{Q}}{2}} = \frac{1}{\sqrt{2}} \left( \mathbf{b}_A + i\mathbf{b}_B \right), \qquad \mathbf{b}_{-\frac{\mathbf{Q}}{2}} = \frac{1}{\sqrt{2}} \left( \mathbf{b}_A - i\mathbf{b}_B \right).$$

Now we can evaluate the nonzero order parameters $\vec{\Psi}_{\boldsymbol{q}}$ and $\vec{\Phi}_{\boldsymbol{q}}$. For the FH case, only $\vec{\Psi}_{\boldsymbol{0}} = \langle \mathbf{b}_A^\dagger \vec{\sigma} \mathbf{b}_A \rangle$ is nonzero, as expected. The nonzero contributions for 2SL order are:

$$\vec{\Psi}^{2SL} = \mathrm{Re}\langle \mathbf{b}_{\mathbf{Q}}^\dagger \vec{\sigma} \mathbf{b}_{\mathbf{0}} \rangle,$$

$$\mathrm{Re}\vec{\Phi}^{2SL} = t_f \left( \langle \mathbf{b}_{\mathbf{0}}^\dagger \vec{\sigma} \mathbf{b}_{\mathbf{0}} \rangle - \langle \mathbf{b}_{\mathbf{Q}}^\dagger \vec{\sigma} \mathbf{b}_{\mathbf{Q}} \rangle \right),$$

$$\mathrm{Im}\vec{\Phi}^{2SL} = t_f \mathrm{Im}\langle \mathbf{b}_{\mathbf{Q}}^\dagger \vec{\sigma} \mathbf{b}_{\mathbf{0}} \rangle, \tag{30}$$

which are equivalent to real space $\vec{\Psi} = \langle \mathbf{b}_A^\dagger \vec{\sigma} \mathbf{b}_A \rangle - \langle \mathbf{b}_B^\dagger \vec{\sigma} \mathbf{b}_B \rangle$, $\mathrm{Re}\vec{\Phi} = \mathrm{Re}\langle \mathbf{b}_A^\dagger \vec{\sigma} \mathbf{b}_B \rangle$ and $\mathrm{Im}\vec{\Phi} = \mathrm{Im}\langle \mathbf{b}_A^\dagger \vec{\sigma} \mathbf{b}_B \rangle$, respectively.

The 4SL order parameters are instead:

$$\vec{\Psi}^{4SL} = \mathrm{Re}\langle \mathbf{b}_{-\frac{\mathbf{Q}}{2}}^\dagger \vec{\sigma} \mathbf{b}_{\frac{\mathbf{Q}}{2}} \rangle,$$

$$\mathrm{Re}\vec{\Phi}^{4SL} = t_f \mathrm{Im}\langle \mathbf{b}_{\frac{\mathbf{Q}}{2}}^\dagger \vec{\sigma} \mathbf{b}_{\frac{\mathbf{Q}}{2}} \rangle,$$

$$\mathrm{Im}\vec{\Phi}^{4SL} = t_f \mathrm{Im}\langle \mathbf{b}_{-\frac{\mathbf{Q}}{2}}^\dagger \vec{\sigma} \mathbf{b}_{\frac{\mathbf{Q}}{2}} \rangle, \tag{31}$$

which again correspond exactly to the real space $\vec{\Psi} = \langle \mathbf{b}_A^\dagger \vec{\sigma} \mathbf{b}_A \rangle - \langle \mathbf{b}_B^\dagger \vec{\sigma} \mathbf{b}_B \rangle$, $\text{Re}\vec{\Phi} = \text{Re}\langle \mathbf{b}_A^\dagger \vec{\sigma} \mathbf{b}_B \rangle$ and $\text{Im}\vec{\Phi} = \text{Im}\langle \mathbf{b}_A^\dagger \vec{\sigma} \mathbf{b}_B \rangle$, respectively. Thus, the real space and momentum space analyses agree.

## B Composite order parameters

The complications of tetragonal symmetry make it difficult to relate $\vec{\Psi}$ and $\vec{\Phi}$ to the more familiar composite order parameter discussed in the $SU(2)$ two-channel Kondo model [46], as in tetragonal symmetry the spin and channel $SU(2)$'s are entangled. Here, we turn to the simpler two-channel Kondo model, valid for cubic symmetry, and discuss the composite forms of $\vec{\Psi}$ and $\vec{\Phi}$. The two-channel Kondo-Heisenberg model is [50],

$$H = \sum_{k\sigma\alpha} \epsilon_{\mathbf{k}} c_{\mathbf{k}\sigma\alpha}^\dagger c_{\mathbf{k}\sigma\alpha} + J_K \sum_{j\sigma\alpha\beta} c_{j\sigma\alpha}^\dagger \vec{\tau}_{\alpha\beta} c_{j\sigma\beta} \cdot \vec{\tau}_{f,j} + J_H \sum_{\langle ij \rangle} \vec{\tau}_{f,i} \cdot \vec{\tau}_{f,j} \tag{32}$$

Here, $\vec{\tau}_f$ are the local pseudospin moments (in cubic symmetry these originate from the $\Gamma_3$ non-Kramers doublet and are mainly quadrupolar), with the pseudospin degrees of freedom labeled by $\alpha, \beta = \pm$. The channel degrees of freedom are labeled by $\sigma = \uparrow$, $\downarrow$ and channel moments will be labeled by $\vec{\sigma}$. The local moments may be represented by $\vec{\tau}_f = \frac{1}{2} f_\alpha^\dagger \vec{\tau}_{\alpha\beta} f_\beta$ (introducing Einstein summation notation), which leads to a quartic Hamiltonian. In the $SU(N)$ large-$N$ limit, this Hamiltonian can be decoupled by two kinds of Hubbard-Stratonovich fields, $b_{j\sigma} \propto \langle f_{j\alpha}^\dagger c_{j\sigma\alpha} \rangle$ and $t_{f,ij} \propto \langle f_{i\alpha}^\dagger f_{j\alpha} \rangle$. Now, the composite order parameter can be roughly derived as,

$$\begin{aligned}
\Psi_j^a &= \langle : b_{j\sigma}^\dagger \sigma_{\sigma\sigma'}^a b_{j\sigma'} : \rangle \\
&\propto \langle : c_{j\sigma\alpha}^\dagger f_{j\alpha} \sigma_{\sigma\sigma'}^a f_{j\beta}^\dagger c_{j\sigma'\beta} : \rangle \\
&\propto \langle : c_{j\sigma\alpha}^\dagger \sigma_{\sigma\sigma'}^a c_{j\sigma'\beta} f_{j\beta}^\dagger f_{j\alpha} : \rangle \\
&\propto \langle : c_{j\sigma}^\dagger \sigma_{\sigma\sigma'}^a \vec{\tau} c_{j\sigma'} \cdot \vec{\tau}_{f,j} : \rangle.
\end{aligned} \tag{33}$$

Here, we are evaluating the normal ordered operator, $\langle : O : \rangle$, and in the last line we recombine $f_\beta^\dagger f_\alpha = \vec{\tau}_f \cdot \vec{\tau}_{\beta\alpha}$, use $\text{Tr}\tau^a\tau^b = 2\delta^{ab}$ and suppress the $\alpha, \beta$ indices in the on-site conduction electron spin density. Here, we have recovered the familiar composite order parameter [46], which is interpreted as a channel dependent Kondo singlet - e.g. - the conduction electron pseudospin density screens the local moment, but has a leftover $SU(2)$ channel degree of freedom. Now we can perform the same procedure for $\vec{\Phi}$.

$$\begin{aligned}
\Phi_{ij}^a &= \langle : t_{f,ij} b_{i\sigma}^\dagger \vec{\sigma}_{\sigma\sigma'} b_{j\sigma'} : \rangle \\
&\propto \langle : f_{i\alpha}^\dagger f_{j\alpha} c_{i\sigma\beta}^\dagger f_{i\beta} \sigma_{\sigma\sigma'}^a f_{j\gamma}^\dagger c_{j\sigma'\gamma} : \rangle \\
&\propto \langle : f_{i\alpha}^\dagger f_{i\beta} c_{i\sigma\beta}^\dagger \sigma_{\sigma\sigma'}^a c_{j\sigma'\gamma} f_{j\gamma}^\dagger f_{j\alpha} : \rangle \\
&\propto i \langle : \vec{\tau}_{f,i} \cdot \left( c_{i\sigma}^\dagger \sigma_{\sigma\sigma'}^a \vec{\tau} c_{j\sigma'} \times \vec{\tau}_{f,j} \right) : \rangle
\end{aligned} \tag{34}$$

We now use $\text{Tr}\tau^a\tau^b\tau^c = 2i\epsilon^{abc}$. The relevant conduction electron quantity is now a pseudospin dependent hopping term that forms a triple product with the local pseudospin moments at each site.

## C Secondary order parameters and moments

The primary order parameters ($\vec{\Psi}$ and $\vec{\Phi}$) and the associated broken symmetries and moments were discussed in the main text. Here we report the secondary order parameters coming from $\vec{\Psi} \otimes \vec{\Psi}$, $\vec{\Phi} \otimes \vec{\Psi}$ and $\vec{\Phi} \otimes \vec{\Phi}$, which could be used to construct a more general AFH free energy. Table 2 contains all of the secondary order parameters, their symmetries and associated moments, for the FH, simple AFH ($t_f = 0$), 2SL and 4SL phases. There are no new symmetries that can be constructed from three or more order parameters. Note that $\vec{\Phi}$ is expected to be smaller than $\vec{\Psi}$ by a factor of $t_f/D$, which also suppresses the secondary moments and makes them more difficult to detect.

$\vec{\Psi} \otimes \vec{\Psi}$ contains only non-time-reversal symmetry breaking electric quadrupole moments that capture the broken tetragonal symmetry, and have been substantially discussed in the main text.

$\vec{\Psi} \otimes \vec{\Phi}$ is substantially redundant with $\vec{\Phi}$ itself, however, in the $XY$ AFH phases, there are additional moments. Particularly, for the 2SL XY phase, the uniform time-reversal symmetry breaking is additionally indicated by $m\Gamma_1^+$ (a dotriacontapolar order parameter) and tetragonal symmetry breaking octupolar moments, $m\Gamma_{3,4}^+$. For the 4SL XY phase, the inversion symmetry breaking is indicated by staggered magnetic quadrupoles that both break ($mZ_{3,4}^-$) and preserve ($mZ_1^-$) tetragonal symmetry. These higher order moments might be useful signatures of the inversion symmetry breaking, as the uniform electric dipole moments, $p_z$ of the phase will be screened in a metal.

$\vec{\Phi} \otimes \vec{\Phi}$ contains the smallest secondary order parameters, suppressed by $(t_f/D)^2$, but are particularly interesting in the 2SL/4SL Z phases, which would correspond to the large moment antiferromagnet. Here, the $\vec{\Phi} \otimes \vec{\Phi}$ moments break tetragonal symmetry even though $\vec{\Psi}$ does not, with the same uniform quadrupole moments as contained in $\vec{\Psi} \otimes \vec{\Psi}$ for the XY phases. Note that these moments are also susceptible to the Imry-Ma disordering mechanism, and are unlikely to be detectable.

When considering moment directions, it is useful to keep in mind that in the large-$N$, mean-field limit, the angular dependence of the order parameters can be written explicitly as:

$$\vec{\Psi} = |b|^2 (\sin\theta\cos\phi, \sin\theta\sin\phi, \cos\theta)$$
$$\mathrm{Re}\vec{\Phi} = t_f |b|^2 (\cos\theta\cos\phi, \cos\theta\sin\phi, -\sin\theta)$$
$$\mathrm{Im}\vec{\Phi} = t_f |b|^2 (\sin\phi, -\cos\phi, 0). \tag{35}$$

These are mutually orthogonal, although this condition may be relaxed for finite $N$, where they may also develop at different temperatures. The triple product of these is an angle-independent scalar, and there are in fact two distinct third order invariants ($\Gamma_1^+$) that can be constructed in tetragonal symmetry: $\Psi_z(\mathrm{Re}\vec{\Phi}_\perp \times \mathrm{Im}\vec{\Phi})_z$ and $\mathrm{Re}\Phi_z(\vec{\Psi}_\perp \times \mathrm{Im}\vec{\Phi})_z$, which are relevant for either the Z or XY phases respectively. These third order terms cause the hastatic order transition to be first order at the multicritical point, but do not generically lead to first order transitions and have no other qualitative effects.

## D Specific heat jump upon transition to region IV

In the main text, we mainly consider the signatures of $T_\Psi$ and neglect signatures at $T_\Phi$. Here we justify this. If we assume $T_\Psi > T_\Phi$, the second specific heat jump associated with $T_\Phi$ is expected to be reduced by at least a factor of $(t_f/D)^2$, as $\Delta C_V = \alpha_{\Psi,\Phi}^2 T_{HO}/u_{\Psi,\Phi} = \alpha_{\Psi,\Phi}(\Psi,\Phi)_{T=0}^2$, and we know $\Phi_{T=0} \approx \frac{t_f}{D}\Psi_{T=0}$. We roughly estimate

Table 2: Possible hastatic phases, their symmetries and their moments. Order parameter (OP) irreducible representations (irreps) are labeled as $\Gamma(Z)_i$ corresponding to particular symmetry under spatial rotations, with $\Gamma$ and $Z$ giving the behavior under translation ($\Gamma$ is uniform, $Z$ staggered with $\mathbf{Q} = Z = [001]$). In addition, the prefix $m$ is used to denote time-reversal breaking, while the superscript $+(-)$ is used to denote parity under inversion. Only the symmetry breaking order parameters and moments are shown ($\Gamma_1+$ are dropped). For each order parameter irrep, only the lowest order multipolar moments are indicated. Magnetic dipolar moments are labeled by $\vec{m}_\perp$ (in plane) and $m_z$ (along $z$-axis), electric dipolar by $\vec{p}$, toroidal dipolar by $\vec{\Omega}$, electric quadrupolar by $Q$, magnetic quadrupolar by $M$, magnetic octupolar by $T$ and magnetic dotriacontapolar by $D$. Less common octupoles and dotriacontapoles are defined in terms of angular momentum tensors as $T_{xyz} = \overline{\langle J_x J_y J_z \rangle}$, $T_z^\beta = \langle \overline{(J_x^2 - J_y^2) J_z} \rangle$ and $D_4 = \langle \overline{J_x J_y (J_x^2 - J_y^2) J_z} \rangle$, where the overline indicates symmetrization. The relative sizes of the moments are not denoted, although they are expected to decrease between rows (for details, see the main text of this Appendix).

| | FH | AFH ($t_f = 0$) | 2SL AFH (Z) | 2SL AFH (XY) | 4SL AFH (Z) | 4SL AFH (XY) |
|---|---|---|---|---|---|---|
| **Primary OPs** | | | | | | |
| $\tilde{\Psi}$ | $m\Gamma_2^+ \oplus m\Gamma_5^+$<br>uniform $m_z$ and $\vec{m}_\perp$ | $mZ_2^+ \oplus mZ_5^+$<br>staggered $m_z$ and $\vec{m}_\perp$ | $mZ_2^+$<br>staggered $m_z$ | $mZ_5^+$<br>staggered $\vec{m}_\perp$ | $mZ_2^+$<br>staggered $m_z$ | $mZ_5^+$<br>staggered $\vec{m}_\perp$ |
| $\tilde{\Phi}$ | - | - | $m\Gamma_5^+ \oplus Z_5^+$<br>uniform $\vec{m}_\perp$,<br>staggered $(Q_{xz}, Q_{yz})$ | $m\Gamma_2^+ \oplus Z_5^+$<br>uniform $m_z$,<br>staggered $(Q_{xz}, Q_{yz})$ | $mZ_5^- \oplus \Gamma_5^-$<br>staggered $\vec{\Omega}_\perp$,<br>uniform $\vec{p}_\perp$ | $mZ_2^- \oplus \Gamma_5^-$<br>staggered $\Omega_z$,<br>uniform $\vec{p}_\perp$ |
| **Secondary OPs** | | | | | | |
| $\tilde{\Psi} \otimes \tilde{\Psi}$ | $\Gamma_3^+ \oplus \Gamma_4^+ \oplus \Gamma_5^+$<br>uniform $Q_{x^2-y^2}, Q_{xy}$<br>and $(Q_{xz}, Q_{yz})$ | $\Gamma_3^+ \oplus \Gamma_4^+ \oplus \Gamma_5^+$<br>uniform $Q_{x^2-y^2}, Q_{xy}$<br>and $(Q_{xz}, Q_{yz})$ | - | $\Gamma_3^+ \oplus \Gamma_4^+$<br>uniform $Q_{x^2-y^2}, Q_{xy}$ | - | $\Gamma_3^+ \oplus \Gamma_4^+$<br>uniform $Q_{x^2-y^2}, Q_{xy}$ |
| $\tilde{\Phi} \otimes \tilde{\Psi}$ | - | - | $m\Gamma_5^+ \oplus Z_5^+$<br>uniform $\vec{m}_\perp$<br>staggered $(Q_{xz}, Q_{yz})$ | $Z_5^+ \oplus m\Gamma_1^+ \oplus m\Gamma_2^+$<br>$\oplus m\Gamma_3^+ \oplus m\Gamma_4^+$<br>staggered $(Q_{xz}, Q_{yz})$,<br>uniform $D_4$, $m_z$,<br>$T_{xyz}$, $T_z^\beta$, | $\Gamma_5^- \oplus mZ_5^-$<br>uniform $\vec{p}_\perp$,<br>staggered $\vec{\Omega}_\perp$ | $\Gamma_5^- \oplus mZ_1^- \oplus mZ_2^-$<br>$\oplus mZ_3^- \oplus mZ_4^-$<br>uniform $\vec{p}_\perp$,<br>staggered $M_{z^2}$, $\Omega_z$,<br>$M_{x^2-y^2}$, $M_{xy}$ |
| $\tilde{\Phi} \otimes \tilde{\Phi}$ | - | - | $\Gamma_3^+ \oplus \Gamma_4^+$<br>$\oplus mZ_2^+ \oplus mZ_3^+ \oplus mZ_4^+$<br>uniform $Q_{x^2-y^2}, Q_{xy}$,<br>staggered $m_z, T_{xyz}, T_z^\beta$ | $\Gamma_3^+ \oplus \Gamma_4^+ \oplus mZ_5^+$<br>uniform $Q_{x^2-y^2}, Q_{xy}$,<br>staggered $\vec{m}_\perp$ | $\Gamma_3^+ \oplus \Gamma_4^+$<br>$\oplus mZ_2^+ \oplus mZ_3^+ \oplus mZ_4^+$<br>uniform $Q_{x^2-y^2}, Q_{xy}$,<br>staggered $m_z, T_{xyz}, T_z^\beta$ | $\Gamma_3^+ \oplus \Gamma_4^+ \oplus mZ_5^+$<br>uniform $Q_{x^2-y^2}, Q_{xy}$,<br>staggered $\vec{m}_\perp$ |

$t_f/D \approx T_{HO}/D \approx 1/30$ for URu$_2$Si$_2$, which means $\Delta C_V|_{T_\Phi} \lesssim .001 \Delta C_V|_{T_\Psi} \approx 3$mJ/mol K, within the experimental noise. The moments are also suppressed, but may be detected by more sensitive techniques.

## E Free energy parameter choices and details of the hastatic transitions

In this section we present the details of the free energy used for numerical calculation of phase diagrams and thermodynamic response across the AFH transitions. Particular emphasis is given to the parameter choices made with microscopic results [33] in mind. We also show additional details of the hastatic order transitions, including the field-locking transition.

Here, we repeat the free energy given in the main text:

$$F = F_\Psi + F_R + F_{\Psi-R} + F_{\Psi-h} + F_{R-h}, \tag{36}$$

which governs the response of the order parameter, $\Psi$ and internal strains/ferroquadrupolar order parameters, $R$. The ferroquadrupolar order parameters are proportional to the elastic strain, and we derive them in detail here. The strain components in tetragonal symmetry are $\left[\epsilon_{z^2}, \epsilon_{x^2-y^2}, \epsilon_{xy}, \vec{\epsilon} = (\epsilon_{xz}, \epsilon_{yz})\right]$ which transform as $\Gamma_{1g}(A_{1g}) \oplus \Gamma_{3g}(B_{1g}) \oplus \Gamma_{4g}(B_{2g}) \oplus \Gamma_{5g}(E_g)$ and are described by the Landau free energy,

$$F_{el} = \frac{c_{11}-c_{12}}{2}\epsilon_{x^2-y^2}^2 + c_{66}\epsilon_{xy}^2 + c_{44}|\vec{\epsilon}|^2 - g_3\epsilon_{x^2-y^2}R_{\Gamma_3} - g_4\epsilon_{xy}R_{\Gamma_4} - g_5\vec{\epsilon}\cdot\vec{R}_{\Gamma_5}, \tag{37}$$

where we omit the bulk(volume) terms. After integrating out the strains from the elastic free energy we find the corresponding ferroquadrupolar orders:

$$\epsilon_{x^2-y^2} = \frac{g_3 R_{\Gamma_3}}{c_{11}-c_{12}}, \ \epsilon_{xy} = \frac{g_4 R_{\Gamma_4}}{2c_{66}}, \ \vec{\epsilon} = \frac{g_5 \vec{R}_{\Gamma_5}}{2c_{44}}, \tag{38}$$

which interact with the order parameter, $\vec{\Psi}$ and external field, $\vec{h}$ and are described by the resulting Landau theory,

$$F_\Psi = \alpha_\perp(T - T_c^\perp)|\vec{\Psi}_\perp|^2 + \alpha_z(T - T_c^z)\Psi_z^2 + u_\perp|\vec{\Psi}_\perp|^4$$
$$+ u_z\Psi_z^4 - v_1(\Psi_{\Gamma_4}^2)^2 + v_2\Psi_z^2|\vec{\Psi}_\perp|^2, \tag{39}$$

$$F_R = \sum_{i=3,4,5}\left(\alpha_R^{(i)}R_{\Gamma_i}^2 + u_R^{(i)}R_{\Gamma_i}^4\right), \tag{40}$$

$$F_{\Psi-R} = \gamma_3 R_{\Gamma_3}\Psi_{\Gamma_3}^2 + \gamma_4 R_{\Gamma_4}\Psi_{\Gamma_4}^2 + \gamma_5\Psi_z\vec{\Psi}_\perp \times \vec{R}_{\Gamma_5}. \tag{41}$$

Note that third order terms in $R$ are allowed by symmetry, but we drop them as well as the biquadratic couplings and anisotropic fourth order terms in Eq. (7). Although allowed by symmetry, they do not qualitatively affect the physics of interest. The couplings to external field are,

$$F_{\Psi-h} = u_h^{(1)}h_z^2\Psi_z^2 + u_h^{(2)}|\vec{h}_\perp|^2\Psi_z^2 + u_h^{(3)}h_z^2|\vec{\Psi}_\perp|^2 \tag{42}$$
$$+ u_h^{(4)}|\vec{h}_\perp|^2|\vec{\Psi}_\perp|^2 + v_h^{(3)}h_{\Gamma_3}^2\Psi_{\Gamma_3}^2 + v_h^{(4)}h_{\Gamma_4}^2\Psi_{\Gamma_4}^2,$$

$$F_{R-h} = \gamma_{hR}^{(3)}h_{\Gamma_3}^2R_{\Gamma_3} + \gamma_{hR}^{(4)}h_{\Gamma_4}^2R_{\Gamma_4} + \gamma_{hR}^{(5)}h_z\vec{h}_\perp \times \vec{R}_{5g}. \tag{43}$$

In order to facilitate relevant discussion for the physics of hidden order in $URu_2Si_2$, our parameter choices for numerical optimization of free energy were heavily influenced by microscopic theories [33].

First, we discuss the parameters for $F_\Psi$, as given in Eq. (39). We chose parameters to reproduce the temperature-pressure phase diagram of $URu_2Si_2$, which is one of many microscopic possibilities. However, once chosen, we use these parameters for the rest of our calculations. As such, we fix:

- $\alpha_\perp = \alpha_z = 1$, $u_\perp = 4$ and $u_z = u_\perp + u'(p - p_c)$, with $T_c^\perp = 1$ and $T_c^z = T_c^\perp + \delta(p - p_c')^3$, where $u' = 1.$, $\delta = .1$, $p_c^0 = 2$ and $p_c' = 1.5$. These choices reproduce the pressure dependence of the transition temperatures, with $p_c' < p_c^0$ necessary to reproduce the rightward curvature of the XY/Z transition at higher temperatures. A large value of $v_2 = 12$ ensures that the transition between XY and Z orders is first order, with no coexistence. Microscopically, the pressure dependence can be induced by tuning the ratio $V_6/V_7$, which tunes the relative energy of XY and Z orders.

- $v_1$ tunes the tetragonal symmetry breaking, where $v_1 > 0$ gives the [110] in-plane pinning consistent with experiment. The pinning can be calculated in the microscopic theory, where $\frac{v_1}{4u_\perp} \sim (T_{HO}/D)^2 \sim .001$. However, this small value makes numerical calculations difficult, and so we have chosen the unphysically large value $v_1 = 1$ for convenience.

The parameters for the ferroquadrupolar components were chosen:

- $\alpha_R^{(4)} = .5$ and $\alpha_R^{(3)} = \alpha_R^{(4)} + \delta_R(T - T_R \tan^{-1}[10(p - p_R)])$, where $\delta_R = 0$ except when we considered the independent $\Gamma_3$ orthorhombic order, with $\delta_R = .5$, $p_R = .5 < p_c^0$ and $T_R = 5$, where these parameter choices give a very sharp second order transition to the $R_{\Gamma_3} \neq 0$ order at $p_Q$. Additionally, we took $u_R^{(3,4)} = 16$.

- While we included $\vec{R}_{\Gamma_5}$ for completeness, it is only relevant if there is XY and Z phase coexistence, which is not found experimentally (and rarely found microscopically). As such, we drop these contributions entirely.

- The coupling between strain and $\Psi_{\Gamma_{3,4}}^2$ is given by $\gamma_{3,4}$, which is found microscopically to manifest as tiny quadrupolar moments proportional to $(T_{HO}/D)^2 \sim .001$. As with $v_1$, this small value would make numerical calculations difficult, and so we have chosen $\gamma_{3,4} = -.5$.

Finally, the field coupling parameters were chosen primarily based on the relatively weak coupling of perpendicular compared to $c$-axis fields, as the $c$-axis field splits the non-Kramers doublet linearly, while the perpendicular fields only split it quadratically. Previous microscopic calculations [33] found that the in-plane couplings were suppressed by $(T_{HO}/D)^2$ compared to the out of plane couplings.

- The longitudinal field ($h_z$) coupling to $\Psi_z$, $u_h^{(1)} = 3.3$ is slightly larger than the coupling to $\vec{\Psi}_\perp$, $u_h^{(3)} = 3$. This difference reproduces the experimental phase diagram in $h_z$, where the hidden order phase is favored over the local moment antiferromagnet [66,67]. This phase diagram was also generically obtained in microscopic calculations [41].

- The isotropic in-plane field couplings were both chosen to be an order of magnitude smaller, $u_h^{(2,4)} = .1$. The difference between them is not important for any of the interesting physics. The anisotropic couplings, $v_h^{(3,4)} = -.3$ give a $\mathbb{Z}_2$ pinning of the

hastatic spinor in transverse field, which governs the maximum magnitude of the torque magnetometry. Again, we have chosen these parameters to be an order of magnitude larger than expected from the microscopics for ease of numerical calculations, where based on the torque results [14], we expect $|v_h|/u_h^{(1)} \sim \chi_{xy}/\chi_{zz} \gtrsim .01$. Note that this is likely an overestimate, as the Landau theory is only valid near the transition and the linear component of $\chi_{xy}$ in $(T_{HO} - T)$ should be extracted.

- The coupling of field to ferroquadrupolar orders was fixed to be $\gamma_{hR}^{(3,4)} = -.5$ and has little qualitative effect.

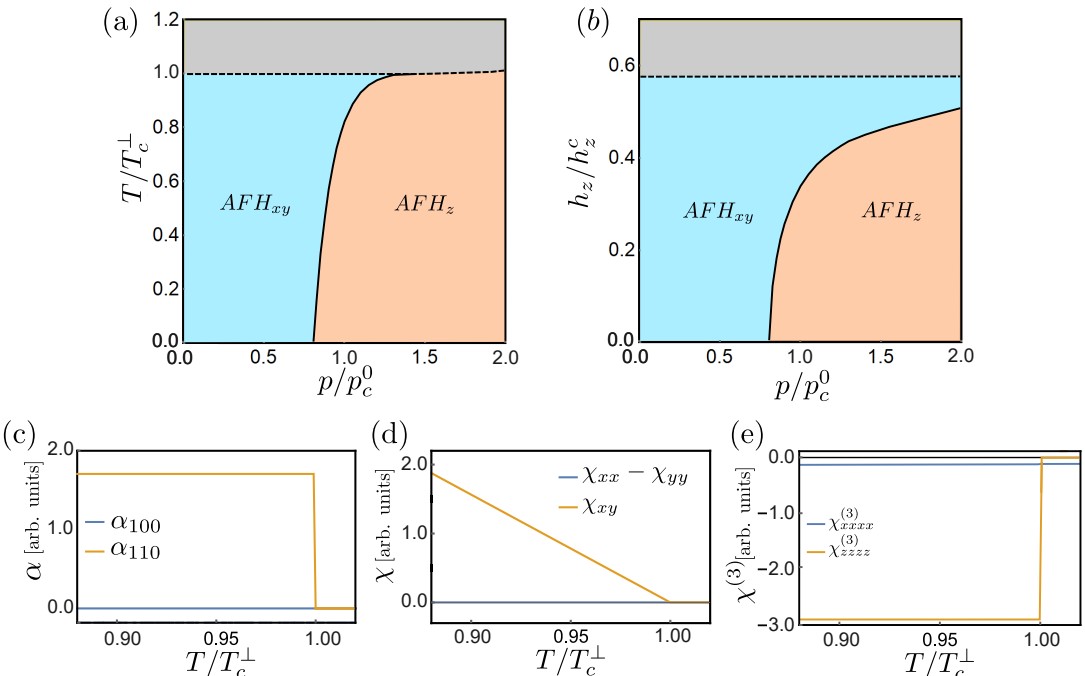

Figure 4: Antiferrohastatic phase diagrams in pressure and $c$-axis field, and thermodynamic responses across the $AFH_{xy}$ transition. (a) Temperature versus pressure phase diagram in zero external field, where the dashed (solid) lines indicate second (first) order phase transitions. The $AFH_{xy}$ phase captures the hidden order behavior, while the $AFH_z$ phase behaves like the local moment antiferromagnet. (b) $h_z$ field suppresses the $c$-axis $AFH_z$ order in favour of $AFH_{xy}$ order. (c) Thermal expansion jumps across $T_c^\perp$. (d) The basal plane susceptibility acquires a linear $\chi_{xy}$ below the transition. (e) The nonlinear susceptibility jumps show a large anisotropy, which is actually expected to be significantly larger for more realistic parameter choices.

The parameters given above reasonably reproduce the experimental phase diagrams in pressure and $c$-axis field, as shown in Fig. 4(a) and (b). We have also explored several characteristic response functions across the hastatic transitions, mainly anisotropic thermal expansion coefficients ($\alpha$) and linear and nonlinear magnetic susceptibility matrix elements ($\chi$ and $\chi^{(3)}$), as well the magnetostriction tensor, but found that magnetostriction involved higher order effects that make it a much less sensitive probe. These response

functions are defined as follows (up to dimensionful constants for the thermal expansion):

$$\alpha_{100} = \frac{1}{L}\frac{\Delta L}{dT}\bigg|_x - \frac{1}{L}\frac{\Delta L}{dT}\bigg|_y \propto \frac{\partial R_{\Gamma_3}}{\partial T}, \tag{44}$$

$$\alpha_{110} = \frac{1}{L}\frac{\Delta L}{dT}\bigg|_{[110]} - \frac{1}{L}\frac{\Delta L}{dT}\bigg|_{[\bar{1}10]} \propto \frac{\partial R_{\Gamma_4}}{\partial T}, \tag{45}$$

$$\chi_{xx} - \chi_{yy} = -\frac{\partial^2 F}{\partial h_x^2} + \frac{\partial^2 F}{\partial h_y^2}, \quad \chi_{xy} = -\frac{\partial^2 F}{\partial h_x \partial h_y}, \tag{46}$$

$$\chi_{xxxx}^{(3)} = -\frac{\partial^4 F}{\partial h_x^4}, \quad \chi_{zzzz}^{(3)} = -\frac{\partial^4 F}{\partial h_z^4}. \tag{47}$$

The thermal expansion coefficients are a proxy for the elastic tetragonal symmetry breaking response and with the [110] pinning of the order parameter, there is a jump in $\alpha_{110}$, as shown in Fig. 4 (c). As already noted in the main text, the jump is difficult to detect, due to $(T_{HO}/D)^2$ suppression of the elastic couplings ($\gamma_{3,4}$) and effects of multiple domains. The tetragonal symmetry breaking is also seen by the onset of $\chi_{xy}$ linear susceptibility, shown in Fig. 4 (d), however torque magnetometry measurements of $\chi_{xy}$ are more subtle and treated in the main text. Finally, the nonlinear susceptibility coefficients are shown in Fig. 4 (e), and exhibit a large Ising anisotropy due to the anisotropic field couplings, as has been observed experimentally [4].

The prediction of a thus far unobserved field-locking transition in large transverse fields provides a key experimental test for hastatic order. In the main text we treat the transition in a simplified model and here we show additional numerical results obtained by the optimization of the full free energy from Eq. (39)-(42) in Fig. 5.

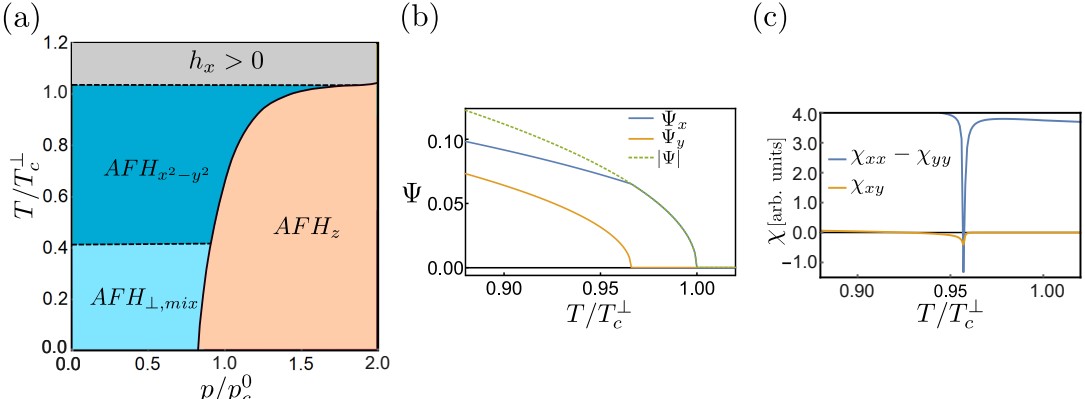

Figure 5: (a) p-T phase diagram in $h_x$ field. The high temperature in-plane phase is field-locked ($AFH_{x^2-y^2}$), thus fully $\Psi_{\Gamma_3}^2$, while the low temperature phase ($AFH_{\perp,mix}$) is characterized by the onset of $\Psi_{\Gamma_4}^2$ component in a second order transition. (b) Order parameter components ($\Psi_x$ and $\Psi_y$) show changes across both primary hastatic and field-locking transitions, while the overall magnitude changes changes significantly only for the main hastatic transition, with important consequences on bulk thermodynamic properties. (c) Susceptibility matrix elements showing divergence across field-locking transition. The change of slope of $\chi_{xx} - \chi_{yy}$ across the main transition is hard to distinguish due to the small size and originally non-zero $\chi_{xx} - \chi_{yy}$ in the presence of external $h_x$ field. Parameters used for obtaining plots are standard quoted in this Appendix.

# F  Discussion of elastic coefficients and resonant ultrasound spectroscopy

Recent resonant ultrasound (RUS) experiments [18] set limits on the existence of two-component order parameters in $URu_2Si_2$ through the absence of observable jumps in symmetry breaking elastic coefficients. In this section, we argue that our two-component order parameter, $\vec{\Psi}_\perp$ lies comfortably within these limits and as such is not excluded as a hidden order candidate by RUS experiments.

As shown in [18], while any OP has jumps in the compressional ($\Gamma_1$) elastic moduli, only multi-component OPs lead to jumps in ($\Gamma_3$, $\Gamma_4$) shear moduli. The magnitudes of elastic moduli jumps ($\Delta c_i$) expected are proportional to the square of OP elastic couplings ($\gamma_{3,4}$ in our theory), more precisely from [18]:

$$\Delta c_{\Gamma_1} \sim \frac{\gamma_1^2}{u_\perp}, \qquad \Delta c_{\Gamma_3} \sim \frac{\gamma_3^2}{v_1}, \qquad \Delta c_{\Gamma_4} \sim \frac{\gamma_4^2}{u_\perp}. \tag{48}$$

The denominator in the expressions above is closely related to the mass of the fluctuation field that couples linearly to the relevant strain component, thus $\Gamma_3$ strain has the weak transverse pinning, $v_1$ in the denominator.

From the microscopic theory, we expect that in the basal plane phases, $\gamma_{3,4}/\gamma_1 \sim (T_{HO}/D)^2$, and $v_1/u_\perp \sim (T_{HO}/D)^2$. While $\Delta c_{\Gamma_4}/\Delta c_{\Gamma_1} \sim (T_{HO}/D)^4$, due to small in-plane pinning, $\Delta c_{\Gamma_3}/\Delta c_{\Gamma_1} \sim (T_{HO}/D)^2$. $\Delta c_{\Gamma_3} = \Delta(c_{11} - c_{12})$ is therefore the largest predicted jump in our theory, but it is still suppressed by $(T_{HO}/D)^2 \sim .001$, corresponding to relative RUS frequency shifts of at best $10^{-8}$, while detected $\Gamma_1$ jumps are $10^{-5} - 10^{-6}$ and the level of noise is at least $10^{-7}$. Thus, even though the hastatic order parameter has multiple components, the weak coupling to the lattice ensures the absence of observable jumps in the shear elastic moduli.

# G  Nematic susceptibility

The nematic susceptibility associated with our two component in-plane order parameter has already been considered by [16], and so here we simply reproduce their calculations and discuss how it applies to our particular system. The nematic order parameter is the electronic manifestation of the broken tetragonal symmetry, and can generically be treated by adding the free energy,

$$F_\mathcal{N} = \frac{a_\mathcal{N}}{2}(T - T_\mathcal{N})\mathcal{N}^2 + \frac{b_\mathcal{N}}{4}\mathcal{N}^4 - \eta\mathcal{N}\epsilon_{\Gamma_4} - \zeta\mathcal{N}\Psi_{\Gamma_4}^2 \tag{49}$$

Here, we have chosen the $\Gamma_4$ nematic order parameter associated with $\Psi_{\Gamma_4}^2$. It has an independent transition temperature, $T_\mathcal{N}$ that arises from fluctuations of $\vec{\Psi}_\perp$; in principle, $T_\mathcal{N}$ can be larger or smaller than $T_{HO}$, but here we assume that it is smaller. The relevant component of the elastoresistivity is proportional to the nematic susceptibility, $\partial\mathcal{N}/\partial\epsilon_{\Gamma_4}$, which contains a jump at $T_{HO}$. The nematic susceptibility is calculated by first solving $\partial F/\partial\Psi_{x,y} = 0$ for $\vec{\Psi}_\perp$ as a function of $\mathcal{N}$ and $\epsilon_{\Gamma_4}$. This solution is inserted into the total free energy, and we then take $\partial F/\partial\mathcal{N} = 0$, and take $\partial/\partial\epsilon_{\Gamma_4}$ implicitly to solve for $\partial\mathcal{N}/\partial\epsilon_{\Gamma_4}$. As in [16], we find,

$$\chi_{nem} = \frac{\partial\mathcal{N}}{\partial\epsilon_{\Gamma_4}} = \begin{cases} \frac{\eta}{a_\mathcal{N}(T-T_\mathcal{N})} & T > T_{HO} \\ \frac{\eta + \frac{2\gamma_4\zeta}{4u_\perp - v_1}}{a_\mathcal{N}(T-T_\mathcal{N}) - \frac{2\zeta^2}{4u_\perp - v_1}} & T = T_{HO}^- \end{cases}. \tag{50}$$

Below $T_{HO}^-$, the nematic order parameter, $\mathcal{N}$ also comes into the denominator and affects the temperature dependence, however, we are mainly interested in the jump. We can use the mean-field specific heat jump results, $\Delta C_V = \alpha_\perp^2/(8u_\perp - 2v_1)T_{HO}$ (or jump in $\partial\Psi_{\Gamma_4}^2/\partial T$ equivalently) to rewrite the jump in the nematic susceptibility to second order in $\gamma_4$ and $\zeta$ as,

$$\Delta\chi_{nem} = \frac{4\Delta C_V}{\alpha_\perp^2 T_{HO}a_\mathcal{N}(T - T_\mathcal{N})}\left[\gamma_4\zeta + \frac{\eta\zeta^2}{a_\mathcal{N}(T - T_\mathcal{N})}\right] + \mathcal{O}\left(\gamma_4^2\zeta^2, \zeta^3\right). \qquad (51)$$

While the microscopics suggested that $\gamma_4$ is suppressed by $(T_{HO}/D)^2$, the electronic nematic order parameter associated with $\vec{\Psi}_\perp$ is expected to be of order one, as estimated by the tetragonal symmetry breaking distortion of the Fermi surface [33, 41]. Therefore the second term gives a significant jump in the elastoresistivity at $T_{HO}$ that is significantly enhanced if $T_\mathcal{N}$ is close to $T_{HO}$. Note that this analysis is only for the main transition; the behavior will be different at the field-locking transition, for example, where the jump is no longer proportional to the (tiny) specific heat jump, but it is still related to the behaviour of $\partial\Psi_{\Gamma_i}^2/\partial T$.

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
