# Peer review of "Signatures of spinorial order in URu2Si2: Landau-Ginzburg theory of hastatic order"

_SciPost Physics_

## Round 1 · Referee Report · Anonymous (Referee 1) · 2021-11-10

Strengths

1) Relevance for the widely studied "hidden order" in URu2Si2. 2) Link to experiments. 3) A intriguing underlying concept.

Weaknesses

1) Some of the basic concepts and definitions used in the paper could be explained more clearly. 2) The main topic of the paper (physical consequences of spinorial order) remains unclear to me. 3) The order parameter which apparently encodes the spinorial nature of the order does not show up in the Ginzburg Landau theory studied mainly in the paper.

Report

The so-called "hidden order" in URu2Si2 has been investigated intensely for more than 30 years. While it has become clear that some type of orbital order is taking place, its precise identification remains controversial. One of the authors suggested in 2013 that a special type of spinor-valued order parameter (called hastatic order) describes the system best and this is one of several follow-up works in this context. One goal of the paper is to provide a Ginzburg-Landau theory in terms of gauge-invariant order parameters and to apply it to the description of URu2Si2. I found that the paper is rather difficult to read and also had problems following the logic of the paper. My main problem with the paper is the following:

The title of the paper includes "Signatures of spinorial order". The author introduces two order parameters psi and phi but apparently all information on the spinoral part are contained in phi ("Phi ... isolates the signatures of the underlying spinorial nature",, The original hastatic proposal [31] has only psi, and does not actually break double-time-reversal symmetry.") According to the title, I expected therefore a theory for phi. To my suprise, the Ginzburg Landau theory is only given for psi (in contrast to what is suggested by Fig. 1c). Perhaps I missed something but I got the impression that the paper does therefore not address its main topic up to some small remarks here and there (including App. C). And the experiments are mainly discussed in view of psi and not phi to my understanding. Is there a phase transition in URuSi where also phi is activated? If not, should one use the term hastatic order at all?

Further remarks: 1) It would have been useful to describe how theta^2 (double time-reversal) acts (i) within the original mean-field theory and (ii) on the two gauge-invarinant order parameters. Even on the level of the gauge theory it is not clear to me whether theta^2 is really broken or whether it is just implemented projectively (i.e., one has to do a gauge transformation to restore the original Hamiltonian). It appears to me that the language of projective symmetry groups (PSG) should be used in this context. In any case, the paper should answer the question posed in the title what the real signatures of the spinorial nature of the order parameter are and whether theta^2 is really broken (and how this is seen experimentally).

2) Presentation: the paper assumes a lot of knowledge from the reader and many issues remain unclear (at least without reading previous papers of the authors). To give a few examples: It does, e.g., not explain what double time-reversal symmetry is (I had to look this up in a previous publication). What is taken from experiment and why ("We know that Tc^perp=Tc^z").? Or what is the precise symmetry group of URuSi (do we have to know only that it is a tetragonal one), which irreps are labeled by 2 and 5.

---

## Editorial Decision

awaiting_resubmission